# Case Series of Acute Meconium Peritonitis Secondary to Perforation of the Ileum in the Antepartum Period

**DOI:** 10.3390/jcm11237127

**Published:** 2022-11-30

**Authors:** Maria Grazia Piccioni, Lucia Merlino, Giulia D’Ovidio, Federica Del Prete, Valerio Galli, Lucia Petrivelli, Flaminia Vena, Valentina D’Ambrosio, Antonella Giancotti, Roberto Brunelli

**Affiliations:** Department of Maternal, Infantile and Urological Sciences, University of Rome La Sapienza, Viale del Policlinico, 155, 00161 Rome, Italy

**Keywords:** abdominal diseases, prenatal diagnosis, fetal malformation, prenatal ultrasound

## Abstract

Perforation of the ileum in the antepartum period resulting in meconial peritonitis is a condition that, although rare, is burdened by several complications. In 80–90% of cases, meconial ileus is the first manifestation of a disease, cystic fibrosis. In the remaining 10–20% of cases, it is caused by other situations, such as prematurity. In most cases, the diagnosis of meconial ileus occurs after birth, although in some cases it can be suspected prenatally, with the finding of a hyperechoic intestine on second trimester ultrasound. The prognosis depends on the gestational age, the location of the obstruction and the presence of fetal abnormalities. Mortality is very high and the recovery of intestinal function in the postoperative course is very high risk. In this case series, we describe two meconial peritonitis and our experience at the center.

## 1. Introduction

Meconial peritonitis is a rare condition mostly associated with perforation of the ileum. Sterile meconium leaks into the peritoneal cavity and creates chemical peritonitis. It is estimated that about 1:35,000 newborns suffer from this condition [1,2,3]. Intestinal perforation occurs proximally at some form of obstruction: volvulus, hernia, intussusception, Hirschsprung disease and meconium ileum with or without cystic fibrosis (FC) [4,5,6,7]. The prevalence of cystic fibrosis in fetuses with prenatal diagnosis of intestinal obstruction is about 10–15%. Therefore, DNA studies for cystic fibrosis should be considered and children with meconium peritonitis should be tested for sweat chloride to rule out cystic fibrosis [8,9,10]. The prognosis is related to gestational age at childbirth, the presence of associated abnormalities, and the site of obstruction. Newborns born after 32 weeks with isolated obstruction requiring resection of only short segments of the intestine have a survival rate over 95% [11].

## 2. Materials and Methods

### 2.1. Case 1

A 32-year-old woman in her second pregnancy, gestational age 34 weeks and 6 days, was admitted to the Obstetrics and Gynecology Department of Policlinic Umberto I, Rome, due to gestational diabetes, fetal ascites and polyhydramnios. The patient medical history includes Still disease, a Sleeve gastrectomy, a gastric bypass and a previous cesarean section. Ultrasound examination ruled out major fetal abnormalities and described the presence of organized ascites as an abdominal cyst, associated with peritoneal calcifications and polyhydramnios (Figure 1). The patient performed prenatal investigations for fetal ascites. A DNA test for cystic fibrosis excluded the genetic cause. An infectious screening ruled out an infection with Parvovirus B19, Cytomegalovirus, Toxoplasmosis and Syphilis. Fetal anemia from fetal maternal isoimmunization was ruled out through the indirect Coombs test, with a negative result. An Magnetic Resonance Imaging (MRI) exam showed bowel dilation > 7 mm and absence of meconium distally to obstruction (Figure 2). Cesarean section was performed at 35 weeks of gestation for non-reacting non-stress test (NST). A 2600 g female newborn was delivered, Apgar score 4-6-9.

### 2.2. Case 2

A 28-year-old woman in her first pregnancy, gestational age 32 weeks and 5 days, complaining with lumbar pain was examined in Obstetrics Emergency Room of Policlinic Umberto I, Rome. Preterm labor and urinary tract diseases were excluded. During obstetric ultrasound fetal ascites was described and a diagnosis of meconium peritonitis was considered. The patient did not undergo any prenatal screening nor second trimester ultrasound. TORCH (Toxoplasma, Others, Rosolia, Cytomegalovirus, Herpes simplex) screening ruled out infectious diseases. Fetal ultrasound at GA(Gestational Age) 32 weeks and 6 days showed abdominal meconial pseudocyst with focal hyperechogenic spots (Figure 3a). Fetal abdominal circumference above 95° centile. Amniotic fluid quantity augmented. Reduced fetal movements. Suspicious interventricular defect. Fetal Doppler Velocimetry in the ranges. The fetal echocardiography, performed by a pediatric cardiologist, found a small subaortic minus of uncertain meaning. No pericardial effusion. A neonatal check-up was advised within the first 24–48 h after birth. Computerized analysis during non-stress test showed a reduced short-term variability and no accelerations, compatible with a class III of the ACOG (American College of Obstetricians and Gynecologists) classification [12] (Figure 3b). A single betamethasone cycle was administrated. Cesarian section was performed at GA 32 w + 6 d for non-reassuring fetal status and meconial peritonitis. A 2330 g female newborn was delivered, Apgar score 6/8. Abdominal swelling and meconium staining of the amniotic fluid were described.

## 3. Discussion and Conclusions

Although the literature describes cases of meconial peritonitis, the diagnostic and therapeutic path is specific on a case-by-case basis. The etiology of the disease remains in many cases unknown and even in our work hypotheses have been made. Prenatal diagnosis is based on the ultrasound examination of key elements such as dilated bowel, ascites, intra-abdominal calcifications and intra-abdominal meconial pseudocysts [1,13]. The pseudocyst is formed to contain the spread in the abdomen of meconium and avoid severe chemical peritonitis. In addition, the use of MRI can help distinguish meconial pseudocysts from other abdominal cysts [14,15]. In our cases, as a result of intestinal obstruction, a condition of polyhydramnios has developed. Uterine overdistension triggered uterine contractions that were partially controlled with tocolytic therapy. It is necessary in differential diagnosis excluding the infectious causes (the most important Parvovirus B19 and cytomegalovirus), and screening of parents for cystic fibrosis, with F508del as common mutation in all ethnic groups [16,17]. In these cases, prognosis is conditioned by the development of multiorgan insufficiencies (primarily pancreatic insufficiency) and the development of acute or chronic respiratory infections [18]. Periodic checks have made it possible to exclude the evolution of fetal clinical conditions and to reach a gestational period of >34 weeks. On the contrary, childbirth should be considered in case of increased fetal ascites, progressive bowel distension or increased polyhydramnios [19]. After surgery, etiological hypotheses were made, identifying vascular dysregulation as the main mechanism. The vascular hypoperfusion of a branch of the upper mesenteric artery may have weakened the intestinal wall, with subsequent reduction of peristalsis, dilation of the proximal intestine and rupture of the wall with meconium leakage in abdomen. No signs of small bowel atresia or volvulus were found [20].

After birth, the fetuses underwent surgery, with partial intestinal resection and anastomosis. In both cases, there was no short bowel syndrome or major complications. From our experience, in the diagnostic suspicion of meconial peritonitis, we suggested transferring the patient to a third level center. A timely diagnosis, with continuous monitoring of fetal well-being, is essential for fetal prognosis. A multidisciplinary work is required throughout the period of pregnancy, to decide on the timing of childbirth and arrange for surgery promptly.

## Figures and Tables

**Figure 1 jcm-11-07127-f001:**
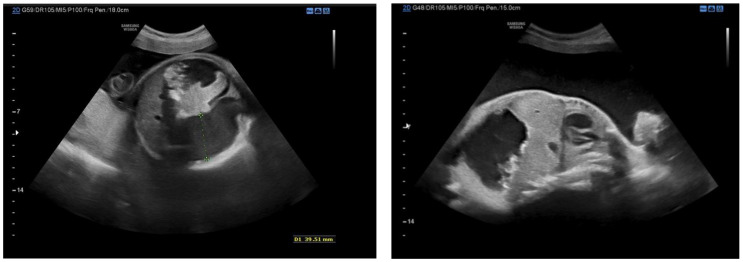
These ultrasound images show one or more ileal loops variously dilated with hyperechoic content (and/or wall). Chemical peritonitis is suspected if intra-abdominal hyperechoic areas are also present. Inflammation of the peritoneum due to perforation of the intestine in the uterus, with meconium transudate into the peritoneal cavity, leading to sterile peritonitis. Intestinal obstruction from adhesions or atresia may occur as a consequence of the inflammatory reaction. In these ultrasound images, an abdominal cyst can be seen, associated with peritoneal calcifications.

**Figure 2 jcm-11-07127-f002:**
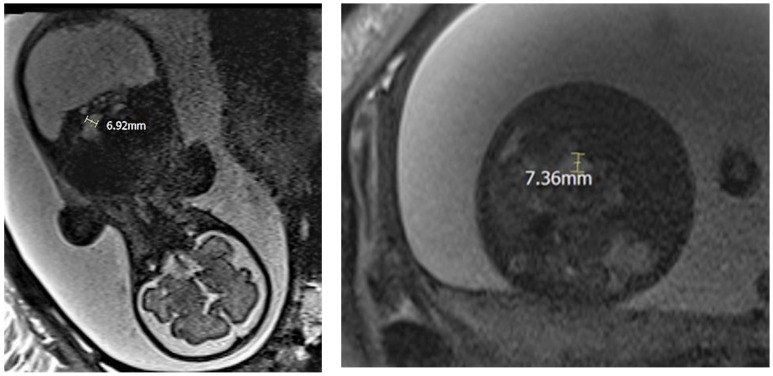
These two MRI images show intestinal dilation > 7 mm and the absence of meconium distal to the obstruction.

**Figure 3 jcm-11-07127-f003:**
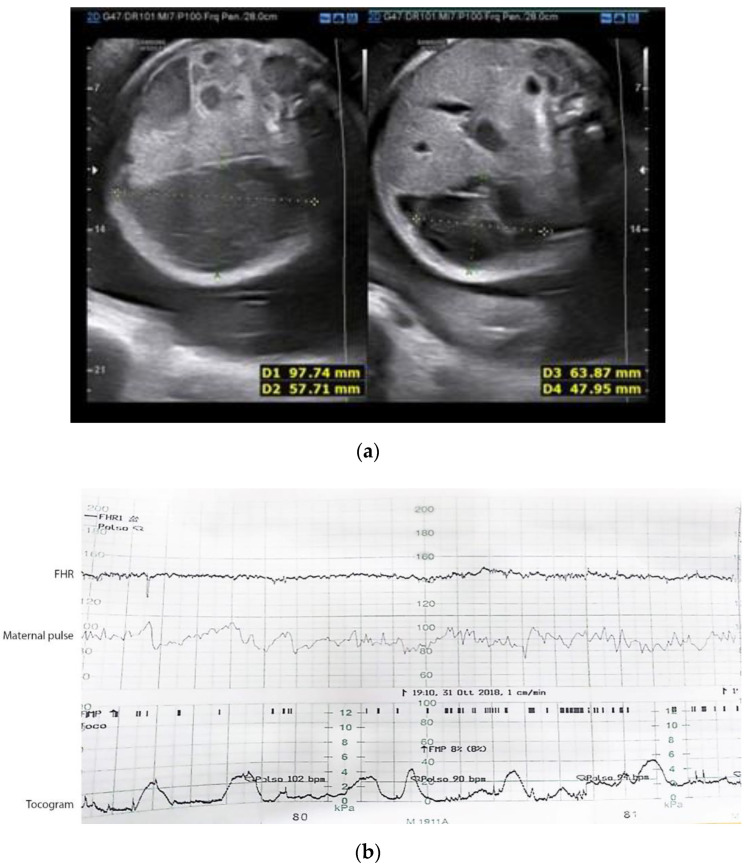
(**a**) Additionally, in this second case, we can observe the typical abdominal cyst, associated with peritoneal calcifications and polyhydramnios, typical characteristics of ileum meconium.; (**b**) Three tracings are shown in this image: the first shows the fetal heart rate ‘FHR’, the second the maternal heart rate and the third the tocogram or uterine contractile activity. In this case, the fetal heart rate shows little variability due to the non-reassuring fetal status. ACOG (American College of Obstetricians and Gynecologists) Classification is Class III.

## Data Availability

Not applicable.

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
