# Peer review of "Case Series of Acute Meconium Peritonitis Secondary to Perforation of the Ileum in the Antepartum Period"

_jcm, 2022, doi:10.3390/jcm11237127_

Round 1

Reviewer 1 Report

Authors report 2 cases of meconium peritonitis. These cases are not really different compared to previous descriptions.

It is not clear why authors chose to mention "ileus volvulus perforation" in the title? In fact there is no peroperative description to validate volvulus. 

Pictures are adequate.

Minor comments:

line 32 I suggest "have survival over 95%"

line 63 "polyhydramnios"

line 78 what means "reduced variability (5-4-3 interbeat variability)"? It would be better to consider short term variability in milliseconds

line 79 "at" instead of "al"

Author Response

1) Q: Authors report 2 cases of meconium peritonitis. These cases are not really different compared to previous descriptions.

1) A: We reported our experience thinking that it could represent a useful contribution for the future to show our therapeutic diagnostic process, also considering the rarity of the two cases described.

2) Q: It is not clear why authors chose to mention "ileus volvulus perforation" in the title? In fact there is no peroperative description to validate volvulus. 

2) A: We changed the title to "Acute meconium peritonitis secondary to perforation of the ileum: a case series"

3) Q: line 32 I suggest "have survival over 95%"

3) A: We changed it in the text

4) Q: line 63 "polyhydramnios"

4)A: We changed it in the text

5) Q: line 78 what means "reduced variability (5-4-3 interbeat variability)"? It would be better to consider short term variability in milliseconds

5) A: In the text we meant the short-term variability (STV), calculated by performing a non-stress test (NST).

6) Q: line 79 "at" instead of "al"

6) A: We changed it in the text

Reviewer 2 Report

Dear Authors,

Interesting paper of high clinical significance.

Author Response

1) Q: Interesting paper of high clinical significance.

1)A: We really appreciate your comment

Reviewer 3 Report

The two case report of meconium peritonitis. I understand the desire to show many pictures, but they should be carefully selected to represent the features in a straightforward manner. There are too many. Also, the quality of the pictures and the resolution of the CTG is poor. They should be digitally processed and submitted as appropriate for publication in the paper.

Case1  In line 46, author described `MRI exam showed bowel dilation > 7mm 46 and absence of meconium distally to obstruction`. This is the key to diagnosis. But the authors did not presented that image.   Case2  As is true in case 1, I do not understand why a supraperitoneal examination has not been performed. The reason is that finding squamous cells in the ascites would strongly suggest a fetid peritonitis.

Author Response

1) Q: The two case report of meconium peritonitis. I understand the desire to show many pictures, but they should be carefully selected to represent the features in a straightforward manner. There are too many. Also, the quality of the pictures and the resolution of the CTG is poor. They should be digitally processed and submitted as appropriate for publication in the paper.

1) A: We have reduced the number of images and tried to improve the quality and resolution of the images.

2)Q: Case1  In line 46, author described `MRI exam showed bowel dilation > 7mm 46 and absence of meconium distally to obstruction`. This is the key to diagnosis. But the authors did not presented that image.  

2) A: We have introduced MRI images

3) Q: Case 2  As is true in case 1, I do not understand why a supraperitoneal examination has not been performed. The reason is that finding squamous cells in the ascites would strongly suggest a fetid peritonitis

3) A: A supraperitoneal examination has not been performed because it's an invasive investigation that would not have changed the therapeutic process from an obstetric point of view.

Round 2

Reviewer 1 Report

"SVT 5-4-3 msec" remains inadequate. I suggest to mention  "reduced short-term variability and no accelerations" (with no precision)

Author Response

Q: "SVT 5-4-3 msec" remains inadequate. I suggest to mention  "reduced short-term variability and no accelerations" (with no precision) 

A: Thanks for the correction. We changed it in the text. 

Reviewer 3 Report

All my requests have been addressed and improvements have been acknowledged.

Author Response

Q: All my requests have been addressed and improvements have been acknowledged.

A: Thank you for your help